# Recent Developments in Nanomaterials-Based Drug Delivery and Upgrading Treatment of Cardiovascular Diseases

**DOI:** 10.3390/ijms23031404

**Published:** 2022-01-26

**Authors:** Nura A. Mohamed, Isra Marei, Sergio Crovella, Haissam Abou-Saleh

**Affiliations:** 1Biological Science Program, Department of Biological and Environmental Sciences, College of Arts and Sciences, Qatar University, Doha P.O. Box 2713, Qatar; sgrovella@qu.edu.qa; 2Department of Cardiothoracic Pharmacology, National Heart and Lung Institute, Imperial College, London SW7 2AZ, UK; iym2001@qatar-med.cornell.edu; 3Department of Pharmacology, Weill Cornell Medicine in Qatar, Doha P.O. Box 24144, Qatar; 4Biomedical Research Center (BRC), Qatar University, Doha P.O. Box 2713, Qatar

**Keywords:** nanotechnology, nanomedicine, nanoparticles, metal-organic framework, cardiovascular diseases

## Abstract

Cardiovascular diseases (CVDs) are the leading causes of morbidity and mortality worldwide. However, despite the recent developments in the management of CVDs, the early and long outcomes vary considerably in patients, especially with the current challenges facing the detection and treatment of CVDs. This disparity is due to a lack of advanced diagnostic tools and targeted therapies, requiring innovative and alternative methods. Nanotechnology offers the opportunity to use nanomaterials in improving health and controlling diseases. Notably, nanotechnologies have recognized potential applicability in managing chronic diseases in the past few years, especially cancer and CVDs. Of particular interest is the use of nanoparticles as drug carriers to increase the pharmaco-efficacy and safety of conventional therapies. Different strategies have been proposed to use nanoparticles as drug carriers in CVDs; however, controversies regarding the selection of nanomaterials and nanoformulation are slowing their clinical translation. Therefore, this review focuses on nanotechnology for drug delivery and the application of nanomedicine in CVDs.

## 1. Introduction

The concept of nanotechnology was first introduced to the public in December 1959 when Richard P. Feynman proclaimed, “There is Plenty of Room at the Bottom” [1] at an annual meeting of the American Physical Society. This new concept of nanotechnology gained more attention in 1986 when K. Eric Drexler published his book, Engines of creation: the coming era of nanotechnology [2]. This book further discussed how nanotechnology could be applied in the medical field to be used for drug delivery, diagnostic imaging of diseases, tissue engineering, and gene delivery systems [2]. Nonetheless, the term “nanomedicine” was first disclosed by K. Eric Drexler, Chris Peterson, and Gayle Pergamit in their book, *Unbounding the Future: The Nanotechnology Revolution,* published in 1991 [3]. The term became popular following the book, *Nanomedicine*, by Robert A. Freitas in 1999 [4]. Since then, nanomedicine has been investigated to revolutionize the therapeutic strategies to impacted different healthcare fields [1].

Nanomedicine is defined as “the use of materials, of which at least one of their dimensions that affects their function is in the scale range 1–100 nm, for a specific diagnostic or therapeutic purpose” [5]. This field expands to integrate biology, chemistry, pharmacology, and material sciences. Despite being described in the early 1990s, nanomedicine is still considered a new and rapidly evolving field that has drawn the attention of medical researchers, the biotech industry, engineers, legislatures, and public consideration. By 2009, 200,000 research articles [6] were published on the use of nanomaterials in medicine and their potential to improve drug administration, implants, surgical devices (e.g., nanoneedles, nanoblades), nanorobotics, and nanosystems that combine therapeutics and real-time disease monitoring [6,7]. One of the emerging interests in nanomedicine is their application as drug carriers to improve the pharmaco-efficacy and safety of conventional therapies [8]. In addition, nanotechnology for drug delivery holds the promise to improve the diagnosis and treatment of several diseases such as cardiovascular diseases (CVDs) [9], neurodegenerative diseases, inflammation, diabetes, orthopedic disorders, and cancer [1]. This review focuses on the use of nanotechnology for drug delivery and its application in CVDs.

## 2. Nanoparticles as Drug Carriers

The use of nanoparticles (NPs) as carriers for drugs can offer many advantages over conventional therapies due to their tunable physiochemical and structural properties [10,11]. Loading conventional therapeutics into NPs, combined with the precise design and accurate tuning, can improve pharmacokinetic/pharmacodynamic properties, efficacy, and solubility by changing the properties of the loaded drug in a way that sides with the therapeutic purposes [10]. Such nanoformulations may also increase the drug’s safety and tolerability by reducing organ toxicity and non-selective drug delivery to undesirable sites. They could also be tuned to prevent the pre-activation and consequent early clearance of the drug by the immune system, thus enhancing drug stability. Furthermore, these formulations may promote a selective, guided, and sustained drug release to the affected tissues when labeled with a specific antibody, maintaining a specific and homogenous drug absorption and distribution [12,13].

Developing NPs from biocompatible and biodegradable materials has unceasingly been a major concern to improve drug delivery and bring considerable advances to other biomedical domains such as tissue engineering and biomaterial sciences [12]. Such development imperatively requires interdisciplinary networking involving biologists, chemists, engineers, and clinicians [12]. The careful and precise design of NPs, the choice of their material structure, and the methodology used to load the drugs into these NPs are all crucial aspects that affect the carried drug’s stability, solubility, pharmacokinetics, and tissue distribution [8]. Thus, the following section provides an overview of the currently used material substrates for NPs, the parameters that influence drug delivery, the available methods to load drugs into NPs, their pharmacokinetics, and their interaction with the immune system. Moreover, the optimized protocols employed to achieve targeted drug delivery will be discussed.

### 2.1. Material Substrate

Different materials from organic or inorganic sources are being used to synthesize NPs, resulting in variable biological, toxicological, and chemical properties. Examples include proteins, micelles, liposomes, dendrimers, polymers, lipids, carbohydrates, nanocrystals, nanotubes, metals, and metal oxides, in addition to other inorganic materials. The efficacy of these NPs as drug delivery systems is affected by the properties of these materials and the NPs structures. NPs are composed of a surface layer, a shell, a core and may exhibit different sizes, shapes, and core/shell thicknesses [14]. The size, shape, polarity, and surface topography of NPs can significantly affect the drug’s biodistribution, tissue absorption, cellular uptake, and accumulation. These properties can be tuned in a manner that facilitates the drug to cross biological barriers, increase its bioavailability, and provide a controlled clearance mechanism [15]. These same features can be used to customize treatments to be disease specific.

### 2.2. Size and Shape

Size is an important factor that ensures the safe travel of the nanoparticles in the bloodstream and determines their accumulation sites. Small nanoparticles are more likely to accumulate in leaky and deformed vessels in cancer and many cardiovascular diseases (due to angiogenesis), and they can also extravasate into normal tissues. In contrast, large nanoparticles cannot easily accumulate in leaky vessels or extravasate into the surrounding normal tissues [16]. Once NPs are used up in the body, different body organs clear them out depending on their size. Small NPs with size <10 nm are removed by the kidney, while NPs > 10 nm are removed by the mononuclear-phagocyte system (MPS) [16]. This indicates that, by tuning and controlling NPs size, we can positively act on their uptake, tissue retention, distribution, and clearance from the body. In addition, NP shape influences their interaction in vivo and their body’s retention due to its impact on fluid dynamics [16]. The shape has also been shown to impact cellular uptake and internalization, as well as cytotoxicity [17].

Nanoparticle functionalities can be tailored by altering the size, shape and/or surface chemistry of the nanoparticles. The purpose of functionalizing the nanoparticles it to help them reach the targeted tissue/organ, it also helps in enhancing the cellular uptake of the nanoparticles and their permeability. Choosing the nanoparticles optimal size depends on the specific type and the location of the targeted tissue/organ; this requires additional studies to understand the interactions between the nanoparticle and cells component of that tissue. Analytical models are also used to guide in designing and functionalization nanoparticles; these methods can help choose the NP shape. For instance, a comparison between spherical, cubic, and rod-like gold nanoparticles showed that spherical particles have the highest uptake in terms of weight; however, rod-like nanoparticles were higher in terms of quantity.

Furthermore, the size of the nanoparticle impacts cellular uptake, as it affects the adhesion strength between the nanoparticles and cellular receptors. Through the manipulation of that interaction, we can use the same nanoparticle (with different sizes) in many types of immune regulations. Small nanoparticles were shown to be great conjugates to vaccine antigens, with high cellular uptake rates compared to microsized particles. In addition, sizes between 25 and 40 nm showed high tissue penetration rates, which aid in activating the adaptive immune responses [18].

Variation in nanoparticles sizes can be accompanied with aggregation problems, where small nanoparticles are shown to aggregate into clusters up to several microns in size. Therefore, studying nanoparticles aggregation is crucial in determining the toxicity and nanoparticle risk in biological systems. Moreover, when small nanoparticles accumulate in organs and tissues with altered pH, it was shown that the more acidic the environment is, the greater the nanoparticles’ aggregation tendency. In addition, nanoparticle aggregation can affect the size distribution, which in turn influences the toxicity and fate of the nanoparticles, altering the exposure location pathways [19].

Nanoparticle aggregation was also shown to affect the mode of cellular uptake in turn modifying the subsequent biological responses. Some nanoparticles use the aggregation on the cell membrane as a way to interact with the membrane proteins and to use it as a way to enter the desired cells. This is often seen with small-sized nanoparticles, when these nanoparticles fail to be wrapped by the cell membrane through receptor-mediated endocytosis, in which they often tend to aggregate, forming clusters of larger sizes to help in the internalization process. The rate of aggregation on the cell surface is regulated by the nanoparticles’ size/shape/surface chemistry and by membrane tension. Therefore, nanoparticle aggregation is sometime beneficial, and understanding nanoparticle interactions with the cell membrane is important in the application of drug delivery systems [20,21].

### 2.3. Surface Chemistry

Similar to what is described for size/shape, nanoparticle surface chemistry properties impact their distribution in tissues/organs. Biocompatible nanoparticles often do not generate an immune response; therefore, biocompatibility of the nanoparticles should be first fully assessed, and the interactions between the nanoparticles and the serum proteins should be investigated. The surface chemistry of the nanoparticle not only affects its function, but it also determines its fate and clearance mood [21].

Nanoparticle surface charge, surface chemistry, and surface functional groups are important in determining the interaction between the nanoparticle and the cellular components of a particular tissue or organ. To do so, nanoparticles often come into contact with extracellular fluids (ECFs) (e.g., blood, lymph and interstitial fluid); this interaction forms a layer or shell that surrounds the nanoparticle, called the corona. This layer often contains the following ECFs components, proteins, ions, sugars, and lipids. The function of the corona is to mitigate nanoparticle cytotoxicity and to facilitate its clearance by the immune system. However, recent studies have shown the possibility of tailoring the nanoparticles’ surface chemistry in order to recruit certain components of the ECFs to prolong its half-life, increase its permeability, tailor the immune response to the nanoparticles, and facilitate its cellular uptake [22].

The clearance of NPs from the bloodstream is primarily influenced by their hydrophobic/hydrophilic proprieties. For instance, hydrophobic nanoparticles are easily recognized by the reticuloendothelial system (RES), tagged with opsonin proteins [18] to be eliminated by the monocytes and macrophages and cleared from the body before reaching the affected tissue. Thus, coating them with a hydrophilic layer (e.g., polyethylene glycol (PGE) can mask their hydrophobicity. PEG can be used to coat hydrophobic drugs to improve their solubility, stability, and bioavailability and to enhance their retention by leaky vasculature [23,24]. However, materials used in the synthesis and the surface modification of nanoparticles should be carefully validated for their safety. Despite having Food and Drug Association (FDA) approved PGE nanoformulations, it still has some drawbacks, such as unfavorable physicochemical characteristics due to particle aggregation, post in vivo administration complications such as hypersensitivity reactions, and developing immune tolerance that limits their pharmacological effect [25].

### 2.4. NPs and the Immune System

Since the introduction of NPs as drug carriers and detection agents, concerns about their effect on the immune system have been raised. Historically, concerns were mainly centered on the immunostimulatory effect of the NPs; however, NPs unintended immunosuppressive and anti-inflammatory effects are currently drawing more attention [26]. For that particular purpose, monitoring immunosuppression, immunostimulation, anti-inflammatory effect of nanomaterials, and whether that effect is to the benefit of the studied disease is crucial to assure the nanomaterials safety [26]. Some of the currently existing nanoformulations possess anti-inflammatory effects on their own, as they reduce inflammatory, immune responses, and oxidative stress in diseased tissues [16]. Metals such as iron, gold, silver, copper, zinc oxide, zinc peroxide, magnesium oxide, nickel, selenium, cerium oxide, and titanium dioxide have been endowed to have anti-inflammatory as well as anti-proliferative effects [27]. Of these NPs, iron oxide NPs have drawn attention due to the many therapeutic properties they possess. Iron oxide NPs have been implicated in the treatment of many inflammatory disorders. For example, a study conducted by Saeidienik et al. showed that iron oxide NPs have neuroprotective and anti-inflammatory effects [28]. In this study, treatment of depressed mice with different doses of iron oxide NPs resulted in the attenuation of the depression symptoms through the modulation of neurotransmitters and the anti-inflammatory effects of these NPs [28]. Other NPs can be designed and engineered to specifically target the immune system causing an intended stimulatory or suppressive effect [29,30]. For example, several studies have shown that iron-oxide-containing NPs possess unintended immunosuppressive effects [29,30], which could be used in the field of organ and tissue transplantation to track the distribution of the administered stem cells and to study the viability of implanted cells within scaffolds intern monitoring the transplanted grafts; in addition, if loaded with other immunosuppressive drugs, they can be used to evaluate drug release and distribution in the tissues/scaffolds [31]. Therefore, NPs modulation of the immune responses led to the development of the nano-immunotherapy field [32,33]. This strategy is essentially used in cancer therapy to stimulate or boost the natural defenses of the immune system to recognize and kill cancer cells. Thus, nano-immunotherapy may increase the ability of the body to recognize and destroy abnormal cells (e.g., cancer cells) while protecting the body from the side effects of chemotherapy [32,33].

### 2.5. Drug Loading and Release Pharmacokinetics

Drugs can be loaded into NPs by encapsulation, conjugation, linking, or other techniques [8]. In addition, they can be loaded with more than one drug to enable combined drug therapy. Examples of combined nanotherapies are the micellar nanoparticles loaded with both bortezomib and doxorubicin, which exhibited a synergistic antitumor effect on ovarian cancer [34]. Another example is the use of PLGA-NPs to carry multiple siRNAs either alone or in combination with drugs showed an increase in the tumor’s sensitivity to the treatment [35,36]. The technicality by which drugs are loaded into NPs affects the carried drug’s stability and solubility, as well as its pharmacokinetics and body distribution [8]. The mechanisms controlling the pharmacokinetics of drug-loaded NPs vary from those controlling conventional drugs and biologics. Therefore, understanding these processes is crucial to improving NPs efficacy and in vivo performance [37]. Glassman and Muzykantov have provided a comprehensive review of the differences between the pharmacokinetics of small drug molecules and drug-loaded NPs. In this review, we only highlight some of the key elements involved in NP pharmacokinetics [37].

One key element that affects drug pharmacokinetics is the duration by which the NPs can stay in the body, which varies according to the class of the NPs. Nanomedicine research reveals the involvement of a protein layer, called corona in defining the NPs half-life [24]. Protein corona is responsible for the NPs clearance from the body as it adheres and accumulates on the surface of the NPs. This accumulation acts as a signal and activates immune cells (e.g., macrophages) to clear the NPs [24]. This unwanted mechanism can be prevented by the surface functionalization and coating of the NPs with organic material such as PEG, which can inhibit the NPs early recognition by the immune system.

### 2.6. Surface Modifications for Targeted NPs Delivery

Some NPs have chemical properties that can help to achieve targeted delivery. Targeted delivery depends on the used nanoformulation and comprises either passive and/or active targeting. Passive targeting occurs due to the non-specific accumulations of NPs in the diseased tissues, a phenomenon seen in remodeled and impaired vessels [23]. These nanoparticles have high enhanced permeability and retention (EPR) properties, which help them to accumulate in the diseased tissues passively. This occurs in the damaged areas of the blood vessels, where the microvascular endothelial cell space is thin and damaged, making it easy for the drug-loaded nanoparticles to pass through the vascular wall [38]. Many CVDs such as acute ischemic stroke, myocardial infarction, atherosclerosis, abdominal aortic aneurysm, varicose veins, and hypertension have elevated vascular permeability [39,40]. Impaired or damaged vasculature is characterized by deformed and remodeled blood vessels with poor structural integrity and accumulation of inflammatory and progenitor cells. This results in having increased levels of the inflammatory cytokines such as TNF-α, TGF-β, IL-12, IL-6, IL-1 β, IFN-β, and cell adhesion molecules such as VCAM, ICAM-1, E selectin, and P-selectin [39,40,41]. Unfortunately, passive targeting cannot eliminate the accumulation of the nanocarriers in tissues that generally have fenestrated blood vessels, such as the liver or the spleen [42]. Conversely, active targeting occurs when NPs are tagged to selectively transport therapies to the desired sites. Tagging NPs surface with an endogenous guiding ligand (micro and/or macromolecule such as proteins antibodies, peptides) that are designed to single out diseased cells, tissues, and organs has the potential to increase the drug’s intracellular and tissue accumulation [16,42].

## 3. Nanoimaging

Nanoimaging refers to the use of nanoformulation as detection agents. Some NPs on their own can offer both pharmacological and imaging properties [15]. In nanoimaging, NPs are used for disease detection and monitoring during follow-up. Similar to their effect with drugs, NPs can improve the pharmacokinetics, efficacy, and safety of the loaded detecting agent and, in some cases, add some targeting benefits. In addition, they can provide better contrast compared to the conventional imaging techniques, thus promoting earlier and rapid disease detection [15]. This feature is a turning point in the prognosis of rapidly progressing diseases, such as cancer, making nanoimaging a valuable diagnostic tool [15]. Finding the proper nanoimaging tool is a herculean task that requires full knowledge of the targeted cell type, suitable photosensitive organic dye or radiolabeling the nanomaterial, and detection wavelength. The ability of the NP to accurately select the desired tissue is critical in nanoimaging to avoid non-specific cell binding and, therefore, to avoid false-positive results [10,43]. Examples of nanoimaging tools are the nanoscale crystals, quantum dots that can be attached to proteins to increase their cellular permeability, and they are often used in cancer detection [44].

Another example is the use of polymers to deliver detection agents such as the poly lactic-co-glycolic acid (PLGA), which is favored because of its biocompatibility and biodegradability. In one study, PLGA was used to encapsulate the highly toxic benzophenothiazinium dye EtNBS used in photodynamic therapy. The encapsulation reduced the amount of free EtNBS, which significantly reduced its toxicity [44]. Using nanoimaging-mediated cardiovascular theranosis is one of the recent areas in the CVD detection area. An example of this is the use of a nano-based theranostic approach in reducing the plaque volume. In that context, it was shown that rTPA tagged iron oxide nanoparticles where more efficient in dissolving clots. Once dissolved, it is important to monitor the migration and fate of any clot remains, which is performed using fluorophores coated nanoparticles.

Another example is the use of nanoparticles in the detection and inhibition of angiogenesis, which is an important step in the atherosclerotic plaque formation. To determine the presence of neovascularization ultra-small super paramagnetic iron oxide nanoparticle that can target integrin α_ν_β_3_ receptor have been developed, once iron nanoparticles target the integrin α_ν_β_3_, angiogenesis is determined and quantified using MR. Furthermore, supermagnetic iron nanoparticles tagged with fluorophores were shown to be effective in the detection and destruction of the inflammatory macrophages in atherosclerotic plaques.

Additionally, nanoimaging can also be used in labeling and tracking down the transplanted Stem cells. A study showed that transplanting mesenchymal stem cells can improve the cardiac activity in myocardial infarction (MI) patients. However, once transplanted, it is hard to know the % of the cells that reached the infracted area. To overcome this problem, supermagnetic iron oxide nanoparticles (SPOIN) showed to be effective in marking the transplanted cells, tracking them down which aids in determining how many of these cells will reach the infracted area as well as knowing the location of the others. Moreover, when patients undergo transplant (e.g., heart transplant), often the only way to determine the presence of organ rejection is through repeated biopsies. Recent studies have shown the possibility of using fluorophore-tagged iron-oxide nanoparticles, which were shown to be highly taken up by macrophages, which are found in abundance during organ rejection [45].

### 3.1. Smart NPs (Dynabeads)

The use of NPs in medicine is limited to nanopharmaceuticals and nanoimaging and blood disorders [46]. Smart nanomaterials (or Dynabeads) are used to magnetically activate and sort cells to purify the blood from harmful compounds such as pathogens and toxins, as well as proteins. This procedure is used to replace the ordinary dialysis methods and employs iron oxide or carbon-coated metal NPs. The iron atoms provide the ferromagnetic or super magnetic properties and can be linked to different antibodies, proteins, antibiotics, or other molecules. Once mixed with blood during dialysis, each linked molecule binds to the relevant target (i.e., cell, protein, other blood components). Then, the fluid is subjected to an external magnetic field whereby blood components attached to the Dynabeads will aggregate around the magnetic pole, whereas all the undesired components will pass by [46,47]. Although in the early stages, this technique can decidedly be used for other blood disorders. Other uses of nanomaterials in medicine include nano-nephrology and nanorobots [46,47].

### 3.2. Multifunctional NPs

Multifunctional nanoparticles are defined as “nanoparticles that are capable of accomplishing multiple objectives such as imaging and therapy or performing a single advanced function through the incorporation of multiple functional units” [48]. These NPs are attractive due to their small size, which allows dosage reduction and improves their toxicity profile. These NPs have a large surface-area-to-volume ratio, which increases their solubility and improves their intracellular uptake. They can encapsulate drugs, thus protecting them from external agents. Most importantly, their size can be adjusted such that they readily diffuse through cell membranes and, in some cases, cross the BBB through different uptake mechanisms [49].

Furthermore, these NPs act as therapeutic cargo that allows controlled and sustained release of the therapeutic agent. Different parameters can be activated to ensure the controlled release of a drug, such as temperature, pH level, enzyme activity, magnetic field, etc. One of the most recent trends in NP formulation is the use of DNA as drug delivery systems, where researchers have focused their attention on DNA nanostructures due to their high biocompatibility and the fact that they are less prone to degradation when compared to synthetic NPs [50]. Similar to cancer, CVDs involve vasculature remodeling and lesion development with affected areas of high porosity. Porous vasculature can utilize the enhanced permeability retention (EPR) effect, allowing the nanoformulation to transport drugs to the affected cells. In ischemic tissue injury, the expression of vascular endothelial growth factor (VEGF) induces the breakdown of the endothelial lining by uncoupling endothelial cell-cell junctions, resulting in leaky vasculature. This results in the accumulation of excessive fluid in the affected area as well as the surrounding tissues. Therefore, tagging the nanoformulation with an anti-VEGF antibody might offer actively targeted drug delivery [51,52].

### 3.3. Nanomedicine and the COVID-19 Pandemic

The nanomedicine field has proven its effectiveness and uniqueness during the COVID-19 pandemic. A year after the COVID-19 pandemic, two nanomedicine-based mRNA vaccines have been, developed, tested, validated, and authorized to be used in facing the still ongoing pandemic. The nanotechnology part of the COVID-19 vaccine which consists of the lipid, enabled the formulations of the first two COVID-19 vaccines in a way that advanced the clinical translation of the nanomedicine drug delivery systems from bench to bed use. The quick translation has distinguished the lipid based nanoparticles from the other forms as it showed its readiness to face the clinical challenge of the rapid development of mRNA vaccine [53]. The development of these two vaccines drew the world’s attention to the transformative potential of the RNA-based nanotherapeutics. Especially with recent findings suggesting the ability of RNA-based therapeutics in targeting previously “undruggable” pathways that are involved in the development of many diseases such as CVDs [54].

Thus far, there are three mRNA-nanotherapeutics approved clinically with many more in different clinical phases. Therefore, the further investigation of this field could have significant implications in the future of nanotechnology-based drug and gene delivery. Now more than ever, the successful delivery of advances such as vaccines, mRNA, and nucleic acid material depends on nanomedicine. Furthermore, the lessons learned from the translation of these lipid-based nanoparticles to the clinical phase can be used in forming the foundation for the safety COVID-19 pandemic tremendous scientific achievements caused by the COVID-19 helped in better shaping the translational of nanomedicines applications.

## 4. Approved and Clinically Tested Nanoformulations: State of the Art

Despite the numerous scientific articles published on nanomedicine every year, the number of nanomedicine formulations currently approved for clinical use is approximately one-tenth of what is reported in the scientific literature [49]. Experimental and regulatory barriers contribute to the low number of nanomedical formulations currently approved for clinical use. The main experimental obstacle is the full characterization scheme and extended toxicity profile of the tested NPs. In contrast, the main regulatory barrier is the lack of specified international regulatory guidelines and the need to illustrate the cost-benefit considerations for using nanomedicine formulations. Therefore, our expectations from the nanomedicine field should remain realistic until we fully understand the nanoformulations and their interactions with the human body [15].

Nevertheless, we still need to develop a more “precise” understanding of this field. This involves accurate characterization measures for the existing nanomedicines to understand their intrinsic properties and biological effects fully. This will enable the design of smart novel nanomedicines that are limited to precisely addressing diseases and personalizing nanotherapies.

As of November 2021, 91 clinical trials, including the term “nano,” were listed as “recruiting” or “active”, and 164 trials were under the term completed on ClinicalTrials.gov [55]. A quick screening through the progress in the nanoformulation clinical trials since 2000’ shows a steady increase starting from 2007, with the years 2013–2015 having the highest number of nanoformulation entering clinical trials [15,42]. In addition, a PubMed search of the word nanomedicine returns 37,170 results. Thus far, there are 52 nanomedicine formulations approved by the FDA [6,56,57], and 34 are approved by EMA (with one nanoformulation only approved in the Netherlands) [6,10,57] (see Appendix A and Figure 1) [58,59,60,61,62]. Of these, three have been discontinued, which indicates the high success rate of the approved nanoformulations [6,10,56,57].

Nanomedicine is one of the hottest topics in cancer therapy, with more than 10,296 hits for “oncology” and “nanomedicine”, and 14,896 hits for “nanomedicine” and “cancer” on PubMed, making it the medical field that benefits the most from nanomedicine [9]. Other fields in medicine are picking up with infectious diseases and nanomedicine having 808 hits in PubMed and 84 hits in LitCOVID for COVID19 issues.

Despite having few clinically approved nanoformulations thus far, there is an abundance of available data, experimental research, and clinical trials reflecting the prevalence of this field [58]. Other medical fields should benefit from the knowledge gained from nanomedicine applications in cancer.

## 5. Nanoformulations in Cardiovascular Diseases (CVDs)

CVDs are considered the major cause of disability and the number one cause of death worldwide [58], with 17.7 million deaths recorded in 2015, a figure that is expected to reach 23.6 million in 2030 (WHO 2017) [63]. In Europe, 45% of all deaths are due to CVDs, accounting for 3.9 million deaths/year [9]. In the United Kingdom, CVDs represent the second cause of death in the country, with 2 million people suffering from CVDs [63,64]. CVDs occurrence is either primary due to a defect in the heart, such as congenital heart disease [65], or secondary to other diseases such as diabetes [66] and hypertension [67]. Despite the advances in the treatment and control of CVDs, some diseases are still poorly managed and could benefit from nanomedicine such as hypertension [68], atherosclerosis [69], thrombosis [70], cardiovascular inflammatory disordered such as myocardium and endocarditis [71,72], stroke [68], myocardial infarction (MI) [73], and pulmonary arterial hypertension (PAH) [74,75]. There are many challenges that face CVDs such as: (i) failure in identifying the gaps in areas between CVDs prevention and treatment; (ii) failure to mark the risk factors and to learn how modify them; (iii) failure in identifying and meeting individual needs; (iv) failure in diagnosing CVDs; (v) failure in accessing first line treatments; (vi) failure in identifying and using advanced CVDs treatments; and finally, (vii) failure in providing patients with supportive care [76]. Overcoming these challenges will lead to the development of new technologies, the development of new diagnostic approaches and the use of modern technologies that will help improving CVDs diagnosis and treatment [77]. In terms of the drug delivery systems, there are several grand challenges and hurdles that need to be addressed and overcome to improve the outcomes of patients with CVD. The first challenge is innovation, where scientists need to understand the basic molecular mechanisms that underline the development the CVD. By understanding these mechanisms relevant drug’s delivery options can be selected. Second, different patients might require different delivery mechanisms, which has initiated the personalized therapeutic option. It is now clear more than ever that each patient comes with their individual: (i) risk factors, (ii) genetic background, (iii) lifestyle environment, and (iv) disease burden. Therefore, it is necessary to employ extensive diagnostics such as imaging, DNA sequencing, and proteomics to improve the clinical decision-making process [78].

Third, available CVD pharmacologic therapies are still combined with considerable drawbacks, these are the drug’s low efficacy, side effects, and drug tolerability doses. A possible way to overcome CVDs challenges are the use of the advances made in the nanotechnology field in medicine and implement it in the CVD area. The use of nanoparticles as drug carriers can overcome these challenges, as there are nanoparticles that naturally interact with different biological pathways; others can be modified in terms of their charges, sizes, solubility to accumulate in the desired tissues. They can also be tagged with agents or ligands to make sure that they accumulate in the desired site. Upon accumulation, these nanoparticles will be stimulated by the intrinsic changes that occur at the disease site (e.g., PH) to release their payload a term called Smart Nanoparticles. Other extrinsic stimuli could be applied in other cases such as using ultrasound, light, magnetic or electrical fields directed at a specific region of the body [78]. This has highlighted the need for exploring new therapeutic strategies to overcome the limitations of current CVD conventional therapeutics. Thus, it has become crucial to apply the advances made in nanomedicine to improve CVDs diagnosis (e.g., Ferumoxytol) and treatment. Of relevance to CVDs is that many nanoformulations, such as polymeric NPs and metallic NPs (metal oxides), enable surface modification, allowing the NPs to code with ligands such as antibodies. This modulation improves their biodistribution and provides greater permeability into inflamed tissues [58].

Despite the late start of cardiovascular nanomedicine (CVN), this field is rushing with many promising lab-scale results. There are 1850 hits for “cardiovascular” and “nanomedicine” on PubMed, with studies working to establish innovative solutions for the current CVDs challenges. Many CVN trials remain focused on detecting, monitoring, and treating atherosclerosis; examples are BLAST, NANOM, and Nano-Athero [9]. The use of stimuli-responsive NPs has also been reported [9]. These NPs are tailored to respond to internal vascular changes such as shear stress, with others responding to external stimuli such as magnetic and temperature-sensitive NPs [9].

### 5.1. Atherosclerosis

Atherosclerosis is a chronic inflammatory disease of the blood vessel walls resulting from the interaction between modified lipoproteins, circulating blood cells, and the endothelium. The etiology of atherosclerosis includes the slow and progressive build-up of vulnerable fatty plaques under the lining of the arterial wall, which gradually narrows the lumen of the artery and restricts blood flow to the heart, impelling myocardial ischemia. Acute myocardial infarction (AMI) is the major complication of atherosclerosis and is defined as a sudden, unpredictable plaque disruption (rupture or erosion) with superimposed thrombus formation. Atherosclerosis is a condition that can primarily benefit from nanomedicine due to the cellular and molecular mechanisms underlying the pathophysiology of the disease [79,80]. Atherosclerosis involves the formation of vascular lesions and the accumulation of inflammatory cells such as macrophages that express and upregulate cell surface receptors (e.g., vascular cell adhesion molecule-1, VCAM-1) that could be used as disease markers [79,80]. Several studies have been published investigating the use of nanoformulations that can target lesions sites and can be taken up by vascular cells through the surface modifying the NPs with either radiolabeled antibodies or fluorescent agents specific for the highly expressed receptors at the lesion site using protease-activatable nanosensors [81]. An example is the use of the surface modified copper sulfide (CuS) NPs to target the Transient Receptor Potential Vanilloid-1 (TRPV1) and reduce vascular lipid accumulation [82]. Elevated levels of blood lipids or hyperlipidemia are well-documented risk factors for atherosclerosis. The incorporation of the chemotherapeutic drug “Paclitaxel” into a cholesterol-rich nanoemulsion (LDE) was found to promote atherosclerosis regression and, when tested in vivo, showed a 65% reduction in the size of the atherosclerotic lesion with low toxicity [83].

### 5.2. Thrombosis

Thrombosis is defined as the formation of a malignant blood clot, and it is considered one of the leading causes of death. Despite knowing that in the process of the thrombus formation, platelets, and coagulation factors play a crucial role, the diagnosis of this disease is often limited to late stages, with treatment options being limited and unable to provide reasonable and effective results. Thus, identifying innovative diagnostic and treatment options are highly recommended in thrombosis [84]. Thrombosis underlines many CVDs, such as pulmonary embolism and myocardial infarction [70,85].

The clinical diagnostic of thrombosis relies on CT imaging, Doppler ultrasound, x-ray, and MRI. These techniques are routinely used in localizing the thrombi. Unfortunately, they fail in providing information about the composition and the age of the clots [86]. This kind of information can be obtained using nanomedicine techniques such as NPs radiolabeled with fibrin ligands and other coagulation factors and components involved in thrombus formation [87,88]. Another example of the use of nanomedicine in thrombosis is the development of the perfluorocarbon-core nanoparticle, which was functionalized by covalent binding to the Phe(D)-Pro-Arg-Chloromethylketone (PPACK) drug. PPACK is a synthetic peptide that can selectively and irreversibly inhibit thrombin activation. However, this drug has a short half-life in the body, and linking it to the perfluorocarbon-core nanoparticle prolonged its presence in circulation and, when tested in vivo, showed significant improvements in antithrombotic activity [89]. Another example is the use of iron oxide nanoworms (NWs) as carriers for a ligand-labeled peptide containing the Thrombin-activatable peptide (TAP); this nanoconjugate was shown to have high selectivity to thrombin [84].

### 5.3. Stroke

Stroke is a cerebrovascular disease that occurs when there is an abnormal cerebral blood flow (CBF), the disturbance then leads to either transient or permanent deficits in the function of one or more parts of the brain. Due to the metabolic and cellular changes caused by the disturbance, cellular death and disruption of the nervous system occurs, and if not intervened in time, the deficit can lead to disability, which is seen in high prevalence in developed countries. Therapeutic options available for stroke include recanalization and then using neuroprotective drugs. Despite the improvement in recovery strategies, there are still some major limitations that dramatically affect patients’ lives in stroke. Some of the main limitations are the narrow therapeutic window, the limited ability to target the brain, and the side effects associated with the used drugs. For these reasons, nanomedicine seems perfect to be used in this area to overcome these limitations and to improve the treatments efficacy. In this area, nanoparticles could be used either as contrast agents or drug carriers [90].

An example of nanomedicine used in treating stroke is the liposome cytidine 5′-diphosphate conjugate tested to treat ischemic stroke. Liposomes and other NPs can be synthesized in a small size that can cross the blood-brain barrier (BBB) without compromising its integrity [91]. In addition, the use of liposome cytidine 5′-diphosphate conjugate can offer neuroprotective capacity in ischemia–reperfusion, which is an effective way of rescuing neural cells and avoiding the brain dead [91]. Another example is the emergence of a new micro-sized particle of the iron oxide (MPIOs) family that contains MRI contrast agents. Using MPIOs conjugated with monoclonal antibodies that targets vascular cell adhesion molecule-1 (VCAM-1) were shown to be effective in defining the “inflammatory penumbra” in ischemic stroke [90].

### 5.4. Myocardial Infarction (MI)

Adult cardiac tissues have limited regenerative capacity that is not enough in repairing the massive loss of heart tissue, which occurs following serious myocardial injuries (ischemia). Delivery of essential therapeutics agents (e.g., growth factors, cells, … etc.) is essential in facilitating the regeneration process. This is where using nanomedicine becomes essential due to the urgent need for the controlled release characteristics nanoparticles have. Applying the advances made in the nanomedicine field to enhance the drug’s cardioprotective potential will revolutionize this area [92]. MI is associated with an inflammatory response, in which neutrophils and macrophages are recruited to the damaged area of the ischemic myocardium. These cells secrete different pro-inflammatory cytokines, chemokines, and proteolytic enzymes such as matrix metalloproteinases and cathepsin B [93]. This has enabled researchers to design probes that can carefully monitor and assess the pro-inflammatory secretome in subjects followed-up, especially for those with high-risk factors. Once fully established, this will help in the early detection of MI [94]. Conversely, iron oxide superparamagnetic nanoparticles possess remarkable magnetic properties that make them promising contrast agents. Furthermore, these NPs are biocompatible, and therefore, they have been used to monitor the homing of stem cells to ischemic myocardium. In addition, they can be used to sort and label stem cells [95]. Another example is the use of VEGF, stromal cell-derived factor-1 (SDF-1), FGF1 and/or Ang-1 loaded in nanoparticles to the ischemic myocardial tissue to stimulate angiogenesis. Recently, nanoparticle delivery through intravenous injection with targeting peptides has merged as a promising strategy. Another study reported an early targeting therapy property when anti-miR-1 antisense oligonucleotide (AMO-1) loaded and myocardium-targeting dendrimer: PEGylated dendrigraft poly-L-lysine with angiotensin II type 1 receptor (AT1-PEG-DGL AMO-1) were used with decreased cell death [92].

### 5.5. Hypertension and Pulmonary Arterial Hypertension (PAH)

Nanomedicine has been applied to improve hypertension treatment. Examples are gold and silica nanoparticles that were developed to enhance the bioavailability of the nitric oxide (NO) supply to diseased vasculature [96]. Another example is cerium dioxide nanoparticles (CeO2 NP) that have antioxidant potential, and when tested in vivo, they showed a significant decrease in both the levels of the antioxidants and the microvascular dysfunction [97]. PAH is a devastating, incurable disease with a poor prognosis and a low survival rate of 2–5 years [98]. Available treatments rely on restoring the normal pulmonary artery pressure by blocking the proliferative/vasoconstriction pathways and enhancing anti-proliferative/vasodilatory pathways [99,100,101,102]. When first introduced, PAH clinical treatment strategies were hampered by the short half-life of the drugs, implying the need for multiple doses or 24/7 infusion. Fortunately, these strategies have been improved to prolong the drug’s half-life in circulation [103,104,105,106,107]. The utmost crucial limitation still facing scientists is the systemic side effects of the current drugs [108,109].

PAH drugs work better in preclinical animal models where PAH is prevented or reversed [110,111,112]. A possible explanation could be that these drugs are given to the tested animals at the early stages of PAH induction; thus, they work in a “prevention” setting, obviously, without predictors. In humans, this type of “pre-treatment” is often not applicable, as it is hard to identify an asymptomatic patient with PAH. Therefore, targeted delivery is of utmost importance to overcome current limitations and improve PAH detection/therapy by taking lessons from the advances made in other CVDs and cancer.

Despite the lack of clinically approved nanoformulations for PAH, preclinical studies are ongoing to investigate their use for this disease. An example is the NF-kappa B polymeric nanoformulation. Studies have shown that in the presence of the PAH lesions, there is an increase in the concentration of the transcription factor NF-kappa B. Blocking the NF-kappa B receptors with the polymeric nanoparticle-NF-kappa B antagonist conjugate can not only prevent PAH development, but can also improve PAH patient survival rate [113]. Most CVDs involve an inflammatory component in their pathogenesis; therefore, using nanoimmunotherapies could improve treatment strategies. A drug class called statins, composed of 3-hydroxy-3-methylglutaryl coenzyme A reductase inhibitor, is commonly used to lower lipid levels, and they have some vasculoprotective effects such as reducing antioxidant properties, inhibiting inflammation, and improving endothelial dysfunction. Statins can prevent inflammation and, therefore, can be utilized in the prevention and in slowing down the development of several CVDs, including PAH. In addition, statin has been incorporated in a polymeric NP to maximize its anti-inflammatory property [69]. Thus far, this application has been tested in various animal models, and its clinical applications are in progress [69]. It is well known that inflammation is involved in the development of PAH [114]. Patients with PAH have high levels of pro-inflammatory cytokines [115] and interferon (IFN) in their plasma [116]. Inflammation also causes the recruitment of immune cells such as macrophages and neutrophils as well as circulating progenitor cells [117]. Accumulated cells will then release an increased amount of cytokines and inflammatory mediators such as IFNs. This accumulation eventually leads to vascular remodeling that affects small pulmonary arteries (~500 mm) [118,119,120] and causes lesions in some cases (the so-called plexiform lesions) [121]. Targeting NPs to lesions and inflammation sites can improve the detection and targeted therapy of PAH and allow studying and characterizing the constituents of the plexiform lesions. Additionally, some NPs can be engineered to have a higher affinity to accumulate at the sites of inflammation and remodeled vessels, a phenomenon well studied in cancer [122,123].

Thus far, there is no drug available to explicitly stop the progressive cellular (inflammatory and progenitor cells) accumulation into the pulmonary artery vessel wall. This pulmonary vascular remodeling is a key pathological feature in PAH, contributing to the progressive narrowing of the lumen responsible for the functional decline and the right ventricle hypertrophy and dysfunction. In PAH, other cells are involved in vascular remodeling, such as the pericytes, which develop into pulmonary artery-like smooth muscle cells suggesting that these cells contribute to the excessive accumulation of pulmonary artery smooth muscle cells and fibroblasts within the pulmonary vascular wall of PAH patients [124]. Pericytes are central regulators of endothelial and smooth muscle cells proliferation, vascular tone, and autoimmunity. Pericytes are responsible for regulating vessel maturation and for stabilization it. Furthermore, an accumulating body of evidence showed that in response to various pro-inflammatory stimuli they can participate in the onset of the innate immune responses. As a response, they then secrete a variety of chemokines and express different adhesion molecules including ICAM-1 and VCAM-1 which then causes the trafficking of the immune cells on the walls of the defected vessel [125,126]. Although pericytes appear to be an attractive therapeutic target in PAH, little is known about the intrinsic characteristics of lung pericytes and their role in the pathogenesis and particularly in the obstructive remodeling in PAH. Therefore, additional studies are needed. Although the contribution of altered immune responses in PAH pathogenesis is consensual, the exact role of inflammation in pulmonary vascular remodeling is still unclear. Whether immune dysregulation represents a cause, or an effect of PAH onset is still unknown [124]. Further studies are needed to assess the inadequate crosstalk between immune mediators and the components of the pulmonary vascular wall to identify novel therapeutical targets [124]. Another NP property that could be used is the affinity of iron nanoparticles to the lung [29], with studies showing the accumulation of the iron nanoparticle MIL-89 in the lung when tested in vivo [29].

Besides benefiting from the passive drug delivery offered by the NPs and the structural changes caused by the PAH prognosis, active targeting can also be achieved by coating the NPs with antibodies against antigens highly expressed in PAH endothelial cells. Examples of these antigens are: Nestin [127], BMPR2; EIF2AK4; TBX4; ATP13A3; GDF2; SOX17; AQP1; ACVRL1; SMAD9; ENG; KCNK3 and CAV1 [128]. A study conducted by Long et al. [129] reported the beneficial effects of BMP9 administration in heterozygous BMPR2 knockout mice, suggesting that compensating for BMPR2 haploinsufficiency by increasing dose of the ligand might constitute a targeted therapy for human PAH [129].

Designing an effective treatment strategy for PAH requires a complete understanding of the cellular mechanisms and the molecular pathways involved in the pathogenesis of the disease. This could explain why combined therapy for PAH has proven to be more effective than single therapy, with the European Society of Cardiology (ESC) and European Resuscitation Council (ERC) guidelines 2015, recommending the use of dual therapy in PAH by using a combination of the Phosphodiesterase type 5 inhibitor (PDE5) and Endothelin-1 (ET-1) drugs as the first-line therapy [130]. Therefore, encapsulating dual or triple PAH drugs might provide a “super” combination of PAH drugs to achieve a higher efficacy. In addition, this encapsulation may also protect the drugs from early immune clearance overcoming the drug tolerance issue seen with PAH drugs. In addition, we can take advantage of the fact that in PAH, endothelial cells are dysfunctional and express specific surface receptors. These receptors could be used as markers in synthesizing NP-coupled antibodies directed against dysfunctional endothelial cells to improve PAH treatment and detection.

The current clinical diagnostic detection for PAH relies on utilizing the suitable heart catheter, an invasive and inconvenient procedure [131]. Furthermore, some NPs, such as Fe-NPs, have shown a specific affinity to the lung, which could be used to direct drugs/agents to the lung for detection and treatment [29]. The ultimate treatment option in PAH is a lung transplant, which is subjected to organ failure [132]. NPs could also be used as a strategy for monitoring lung rejection [29] by detecting the presence of macrophage infiltration and measuring protease activity [75]. Thus, organ rejection could be predicted at the early stages to provide alternative options to save patients’ lives. Despite their promise, all these PAH drugs incorporating NPs have not yet been transitioned to clinical trials [75].

## 6. CVN: Bench to Bed Translation Challenges

Despite having many CVN formulations that are either being used in medicine (Appendix A) or being tested in vitro and in vivo, the effective translation of these nanoformulations is still in process, with most of these formulations struggling to beat the challenges of the preclinical stage as they are hampered by challenges that significantly slows down this process. These challenges include: (i) nanoparticles development and manufacturing, (ii) validating the safety/tolerability of the NPs when intact and when metabolized by the body, (iii) NPs stability in bench and biological environments, (iv) NPs biocompatibility, (v) drug loading and release efficacy and pharmacokinetics, (vi) regulating their pathway requirements and finishing the clinical trials within reasonable time, and finally, (vii) the feasibility of scaling up the produced nanopharmaceuticals [9,78]. However, these challenges are also faced by other cardiovascular drugs that do not have advanced delivery system. Furthermore, these challenges should be met while adhering to the Good Manufacturing Practice, as compliant production is essential [133]. Achieving this will require multi-disciplinary collaborations between research experts, both basic and clinical, academia, and industry to successfully translate NPs with high potential from bench-studied materials to viable clinical products. During the last two years of the still ongoing pandemic, we have seen that a nano-based vaccine rose to the occasion and is now saving the lives of millions of people. This achievement will certainly reshape the way we view, study and implement nanomedicine in many fields, including in the CVDs area [78].

## 7. Conclusions

Nanoformulations represent an emerging tool in biology and medicine. They possess several physicochemical and biological properties attributed to their size, shape, and surface, making them a promising platform for launching alternative non-conventional therapies. We need to use all the advances made in the nanomedicine research field to develop a more “precise” understanding of this field to translate the findings to a clinical scenario. This involves gathering accurate information about the existing nanotherapeutics and thoroughly estimating their intrinsic properties and biological effects. This will enable the design of smart nanoformulations that are created to precisely address diseases and to personalize nanotherapies.

Furthermore, several aspects must be addressed for a nanoformulation to be translated from the laboratory to the drug market. These involve a comprehensive assessment of nanoformulation chemical and biological proprieties, identifying the required ethics and other regulatory guidelines, identifying the market size, costs, and commercial development. A strong network with pharmaceutical companies will allow the researchers to better and rapidly translate the considerable number of preclinical findings into novel drugs for precision medicine. An example of this network has been established during the current pandemic, which has allowed for the fast development of a lipid-based nano-vaccines that have saved the lives of millions. A lesson could be taken from this success to be applied in another field, such as in CVDs.

## Figures and Tables

**Figure 1 ijms-23-01404-f001:**
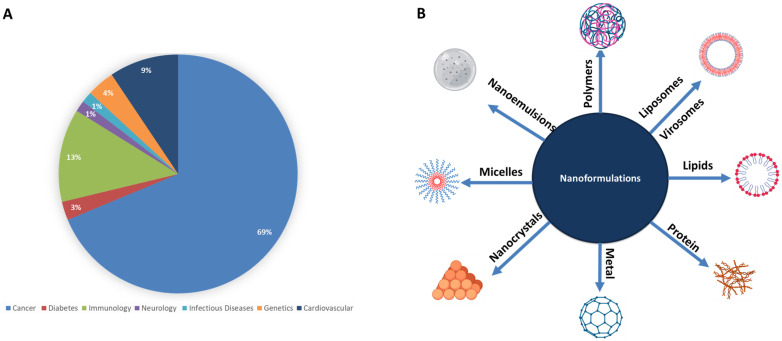
(**A**) Percentage of nanoformulations application in each medical field, and (**B**) different nanoformulations currently in use in the medical field.

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
