# Peer review of "Recent Developments in Nanomaterials-Based Drug Delivery and Upgrading Treatment of Cardiovascular Diseases"

_ijms, 2022, doi:10.3390/ijms23031404_

Round 1

Reviewer 1 Report

General comments:

The title promises a review about nanoparticle mediated drug delivery in CVD. However, it is more a general summary on nanomedicine with less than the expected specific content on drug delivery and even less on CVD related knowledge. The text is unbalanced also inside the CVD part with too many details on PAH and much less content on other aspects. Surprisingly, mRNA containing lipid nanoparticles as vaccines seem to be completely missing from Table 1 and 2. These nanomedicine products should be mentioned, together with nano-enabled RNA based therapies for cardiovascular diseases.  https://pubmed.ncbi.nlm.nih.gov/33709981/

I suggest also to change the title to fit more to the content of the article.

The text is well readable, but it seems to be a bit immature with many typos and errors (see specific comments.)

Discussion on size (Page 3) would deserve mentioning the aggregation of particles and the formation of larger aggregates that might compromise drug safety.

Specific comments:

Page 2 Line 60: …can improve pharmacokinetic/pharmacodynamics…  Suggested: can improve pharmacokinetic/pharmacodynamics properties…

Page 2 Line 91.. that enables the drug to reach inaccessible areas.. Suggestion: that facilitates the drug to cross biological barriers…

Page3 Line 143 coukd  - could

Page4 Line 146 othe…other

Page 4 Line 148: .. is essentially used in cancer to stimulate  Suggested: is essentially used in cancer therapy to stimulate 

Page 4 Line 169 Nanomedicine research reveals the involvement of a protein called corona … Suggested: Nanomedicine research reveals the involvement of a protein layer, called corona …

Page 26: Title of Figures 1 and 2 seems to be exchanged

Page 27 Lines 332 and 352 The full paragraph is repeated

Page 31 Line 544 the effective translational of these nanoformulations is still in process… Suggestion: the effective translation of these nanoformulations is still in process…

Page 31 Line 562: This will enable the design of smart nanoformulations that are limited to precisely addressing diseases and personalizing nanotherapies. 

Meaning of this sentence is not clear. Suggestion: This will enable the design of smart nanoformulations that are created to precisely address diseases and to personalize nanotherapies. (?)

Author Response

We are thankful to the reviewer 1, the commentaries and suggestions have been very useful to rise the quality of our manuscript making it more easily readable.

Please find attached the point to point answers.

Kind regards

Reviewer 2 Report

Reviewer comments:

Title: Nanoparticles-mediated Drug Delivery for Cardiovascular Diseases

ID: ijms-1485980

(a) ARTICLE RANKING

* Good

(b) RECOMMENDATION

* Minor revision.

(c) Comments of Reviewer

TITLE:
* Nanoparticles-mediated Drug Delivery for Cardiovascular Diseases;

Suggested title: “Recent Developments in Nanomaterials-Based Drug Delivery and Upgrading Treatment of Cardiovascular Diseases”

ABSTRACT:

* Describe current challenges in the treatment of CVDs and the need for nanomaterials-based interventions and future implications.  

INTRODUCTION
* It seems too short, bring more connectivity and discuss recent developments in literature.

MAIN TEXT

 * Elaborate text of subsections 2.2.1. Size and Shape, 2.2.2. Surface Chemistry, 3.1. Smart NPs (Dynabeads)

* Elaborate subsection title 3. Nanoimaging

*Make sure paragraphs are made with sufficient length.

Divide Table 1. FDA-approved nanoformulations, their composition, loaded drug, producing company, and approval year; divide the table into three or more depending on need, for example, table for polymer-based Nano formations, metal nanoparticles-based Nano formations, Oxide-nanomaterials-based Nano formulations.

Figure. 1 and 2 is not enough for this review article, there is a need for a few schematic illustrations on Nanoformulation for example Nanoemulsions, Protein-Protein Conjugates, Virosomes, Liposomes,

CONCLUSIONS
* Revise this section with the consistency of the revised manuscript.

Author Response

Kindly find attached the answers to the reviewer's 2 commentaries and suggestions. We wants to sincerely thanks the reviewer for the criticism that allow us to improve the quality of our manuscript.

Kind regards

Round 2

Reviewer 1 Report

Thank you for considering the suggestions.